Experimental assessment of the effects of sustainable landscaping practices on beetles in a new residential land development

Pandolfi Alessandra pndalessandra@gmail.com alessandra.pandolfi@ucf.edu 1
Bohlen Patrick J. 2
Iannone III Basil V. 3
Moffis Brooke L. 4
Skelley Paul E. 5
Jenkins David G. 1
1 Department of Biology, University of Central Florida , Orlando , FL , United States of America
2 Arboretum, University of Wisconsin-Madison , Madison , WI , United States of America
3 School of Forest, Fisheries, and Geomatics Sciences, University of Florida , Gainesville , FL , United States of America
4 UF/IFAS Extension Lake County, University of Florida , Tavares , FL , United States of America
5 Florida State Collection of Arthropods, Florida Department of Agriculture and Consumer Services, Division of Plant Industry , Gainesville , FL , United States of America
Silva Daniel
Electronic publication date: 2025 Dec 16
Publication date: 2025
Volume: 13
Electronic Location ID: e20415
Received 2025 Jun 19; Accepted 2025 Oct 28
Copyright: ©2025 Pandolfi et al.
Copyright year: 2025
Copyright holder: Pandolfi et al.
License: This is an open access article distributed under the terms of the Creative Commons Attribution License, which permits unrestricted use, distribution, reproduction and adaptation in any medium and for any purpose provided that it is properly attributed. For attribution, the original author(s), title, publication source (PeerJ) and either DOI or URL of the article must be cited.
License URL: https://creativecommons.org/licenses/by/4.0/

Keywords: Urban sprawl, Soil amendments, Water conservation, Insect conservation, Taxonomic diversity, Functional traits, Functional diversity

Funding: The Sunbridge Stewardship District (“District”) The UF/IFAS Center for Land Use Efficiency The UF/IFAS Program for Resource Efficient Communities The Florida Chapter of The Nature Conservancy (TNC-FL) through their partnership with UF/IFAS The UCF College of Graduate Studies Open Access Publishing Fund The James and Annie Ying Eminent Scholar in Biology endowment Funding for materials and equipment was provided by the Sunbridge Stewardship District (“District”). Additional support for this work came from the UF/IFAS Center for Land Use Efficiency, the UF/IFAS Program for Resource Efficient Communities, and the Florida Chapter of The Nature Conservancy (TNC-FL) through their partnership with UF/IFAS. The article processing charges were provided in part by the UCF College of Graduate Studies Open Access Publishing Fund and in part by the James and Annie Ying Eminent Scholar in Biology endowment. The funders had no role in study design, data collection and analysis, decision to publish, or preparation of the manuscript.

==============================
As cities sprawl globally, the need for sustainable landscaping practices becomes more critical for addressing issues like water conservation, soil health, and local biodiversity. Beetles are valuable environmental indicators of insect habitat quality and are ecologically important. Here we experimentally assessed the response of ground-active beetle communities to sustainable landscaping practices (i.e., drought-tolerant native plants, reduced irrigation, and compost-amended soils) in an ongoing suburban development in Central Florida, one of the fastest growing regions in the USA. We assessed beetle activity density, diversity, and ecological functionality during the wet and dry seasons using pitfall traps, a well-tested technique for sampling ground arthropods. We hypothesized that the reduced compaction and better moisture retention of compost-amended soil would create better conditions for soil-dwelling beetles that use soil for egg deposition and larval development. We also hypothesized that plant growth-form traits would affect the spatial distribution of beetles within the experimental plots. Finally, we hypothesized that irrigation treatments would not affect beetle communities which are adapted to the region’s seasonal rainfall patterns. Results showed that beetle species richness positively correlated with percent cover of native ground cover forbs, and that beetle activity density tended to decline with increasing pine straw mulch coverage. Warmer soils enhanced both richness and activity density, while both of these metrics exhibited a hump-shaped pattern relative to soil moisture, peaking at intermediate levels. Trait-environment relationships revealed that beetle traits, including diet, body size, and larval substrate use, varied with microclimatic conditions. Trait diversity rose with species richness but plateaued due to trait saturation, while uneven trait distribution suggested filtering and niche separation driven by short-term climatic conditions. This work highlights the importance of research aimed at identifying landscaping approaches that both support insect biodiversity and meet environmental sustainability objectives in urbanizing landscapes.

Introduction

More than half of the world’s eight billion plus people now reside in urban areas—a figure projected to rise to 70% by 2050 (DESA, 2018). Estimates of urban landscape growth vary by data source and country but have roughly tripled between 1985 and 2015 (Chakraborty et al., 2024). At a regional scale, urban sprawl poses significant challenges for biodiversity by habitat change, fragmentation, increased uninhabitable space, and increased mortality risk (Farina, 2022). At a local scale, urbanization alters water availability and soil conditions (McGrane, 2016; Chen et al., 2017). The expansion of impervious surfaces reduces precipitation infiltration and significantly increases runoff (Booth, 1991), while topsoil removal hinders vegetation growth and compromises plant health (Zuazo & Pleguezuelo, 2009). Heavily urbanized areas often experience significant vegetation loss and the homogenization of ecological communities, including insects (Sánchez-Bayo & Wyckhuys, 2019). Beyond these structural changes, ecosystems can also be impacted by invasive species and pests, which further disrupt community dynamics (Schmitz & Simberloff, 1997).

Conventional landscaping practices—often reliant on turf grass and non-native ornamentals—remain widespread, contributing to ecological homogenization and high-water demand. In the U.S., for instance, 30% of household water use is dedicated to outdoor irrigation, rising to 60% in warmer and/or drier regions (Taylor et al., 2021a; Taylor et al., 2021b; U.S. EPA, 2024). In response to recurring droughts, several cities around the world have introduced stringent water-use regulations along with incentive programs to promote sustainable landscaping and improve irrigation efficiency (Kenney, Klein & Clark, 2004; Nagourney, 2015; Borah, 2025). Despite the efforts, poor management of irrigation systems remains a persistent challenge, often undermining water conservation goals. As noted in previous studies, inefficient scheduling is one of the main causes of outdoor water waste, especially in newer residential neighborhoods where sprinkler systems are too imprecise and do not deliver appropriate flow rates to different plant types (Sowby & Lunstad, 2023).

One promising solution to these issues is incorporating more native plants into suburban landscapes. Native plants are adapted to local climate conditions, resulting in lower management needs, such as irrigation demand, compared to the non-native plants frequently used in these settings (Shapiro et al., 2015; Hardberger et al., 2025). In various cities around the world, particularly those affected by extreme temperatures, increased urbanization and climate change have led to water stress and growing interest in xeriscaping limiting turf areas and using grass species better adapted to the region (Chow & Brazel, 2012; Çetin, Mansuroğlu & Onac, 2018). These practices not only support water conservation efforts but also provide essential habitat for native insects and other wildlife, contributing to more resilient and ecologically diverse urban environments (Burghardt, Tallamy & Shriver, 2009; Clem et al., 2021). It has also been suggested that the nursery industry—commonly a major source of non-native plants in urban areas—has both the opportunity and responsibility to support native biodiversity (Perry & Cox, 2024). In some regions, sustainable landscaping practices—including those that incorporate native species to create water-conserving landscapes—are supported by legislation. For example, in Florida, the Florida-Friendly Landscaping Law prohibits local ordinances from banning such practices (Florida Statutes §373.185: https://www.flsenate.gov/Laws/Statutes/2012/373.185).

An additional challenge in urban landscaping is the poor quality of soils (Yang & Zhang, 2015). Before landscaping can be successfully established in newly developed urban areas, soil often requires management to address the degradation caused by construction activities (De Kimpe & Morel, 2000). Common practices such as site grading, top-soil removal for resale, and compaction from heavy machinery result in poor soil structure, reduced organic matter, and limited water-holding capacity (Craul, 1999; Jim, 1998). In many developments, fill soil, which is often subsoil from elsewhere with little to no organic content, is brought in, further reducing soil quality (Gregory et al., 2006). These disturbed soils can compromise plant establishment and performance, including native species that are adapted to local conditions but still require adequate rooting environments and moisture availability.

One strategy for improving sandy or compacted soils is the incorporation of organic amendments such as compost, which reduces bulk density, increases porosity, and enhances water infiltration and retention (Krull, Skjemstad & Baldock, 2004). These improvements can reduce irrigation needs and runoff, promoting healthier plant communities. While native plants are often promoted for their resilience and ecological benefits, recent research has shown that under poor soil conditions, native and non-native species may not differ significantly in performance (Tartaglia & Aronson, 2024), highlighting the importance of addressing soil quality as a foundational component of sustainable landscaping.

Beyond plant health, the quality of urban landscapes also profoundly influences other ecological communities, particularly insects. Recent evidence suggests that insect populations are steadily declining (van der Sluijs, 2020). A recent review of the main drivers of the decline of the entomofauna suggested that among the most diverse insect orders, Coleoptera, Lepidoptera and Hymenoptera, land-use change, and landscape fragmentation are the main cause of decline (Sánchez-Bayo & Wyckhuys, 2019), meaning that urbanization may contribute to this trend. While much research on insect decline has focused on pollinators like bees and butterflies (Zajdel et al., 2025), ground-dwelling insects have received less attention, even though they play a vital role in soil health and nutrient cycling and are effective bioindicators of environmental change (Rainio & Niemelä, 2003; Gerlach, Samways & Pryke, 2013). Among ground-dwelling beetles, carabids have been widely studied, particularly in relation to urbanization gradients (Niemelä et al., 2002; Magura, Tóthmérész & Molnár, 2004). However, far less is known about how urban environmental conditions affect other beetle families and the broader beetle community (New, 2010). Given the role of beetles in maintaining ecological multifunctionality, it is important to expand research beyond carabid beetles to understand the contributions of diverse beetle groups in urban ecosystems. Urban landscaping practices that incorporate drought-tolerant native plants and soil amendments may enhance vegetation structural complexity and increase the abundance and diversity of resources available to soil-surface invertebrates (e.g., Salisbury et al., 2020).

To explore these ecological effects on ground-dwelling beetles, we tested sustainable landscaping strategies—reduced irrigation regimes and soil compost amendments in diverse native plant mixtures—in a field experiment near an emerging residential community in Central Florida, USA. This region exemplifies the ecological challenges of rapid urban growth, as it is among the fastest-growing in the United States (US Census Bureau, 2025) and features a subtropical climate with distinct dry and wet seasons, sandy soils with low water retention, and widespread turfgrass-dominated, water-intensive residential landscaping (Haley, Dukes & Miller, 2007). We evaluated beetle activity density (relative abundance), species diversity, and ecological trait variation in relation to the experimental treatments and native plant species cover. We hypothesized a priori that beetle diversity and activity density would respond positively to compost-amended soils because compost addition can enhance soil quality by reducing compaction, improving soil structure, and better retaining moisture (Fisher, 2023) which would improve habitat conditions for soil-dwelling beetle eggs and larval development and provide resources for the detritivore food web (Pimentel & Warneke, 1989). We also hypothesized that beetle communities would respond to differences in plant cover based on plant traits. Lastly, we hypothesized that irrigation treatments would not affect beetle communities substantially because detected species are adapted to the local climate. By evaluating sustainable landscaping strategies in this rapidly developing region, we aim to inform approaches that mitigate the ecological impacts of residential development and promote biodiversity in human-dominated landscapes.

Materials & Methods

Site description

Florida’s humid subtropical climate and sandy soils with low water-holding capacity present unique challenges for managing urban green spaces. During the wet season (May through October), average temperatures range from 23 °C to 31 °C, while in the dry season (November through April), temperatures range from 15 °C to 24 °C. Monthly rainfall averages about 62.5 mm during the rainy season and 20.2 mm during the dry season. These conditions, combined with rapid urban development and conventional landscaping dominated by turfgrass and ornamental plants, make irrigation essential for residential landscapes (Haley, Dukes & Miller, 2007). In some counties, landscape irrigation accounts for 60–70% of household water use (Taylor et al., 2021a; Taylor et al., 2021b), with common issues including improperly programmed irrigation systems that water too frequently or for too long (Dukes & Cárdenas, 2024; Dukes, Cárdenas & Haman, 2024).

This study was conducted at the Base Camp (welcome center) within the Sunbridge development, situated in southeastern Orange and northern Osceola Counties, near the outskirts of metropolitan Orlando (28.334 N, 81.191 W) (Figs. 1A–1B). Spanning approximately 11,107 hectares, Sunbridge is a large-scale, phased residential and commercial project that is planned to include more than 30,000 new home sites (single and multifamily) built on ∼10,900 ha of former grazing land over the next few decades, alongside designated areas for office, industrial, and retail use. The soils are mostly poorly drained fine sands over a flat topography characterized by many low-lying wetlands. The grazed uplands were originally a mixture of pine flatwoods and wet prairies. The areas immediately east of the development are not developed. Field experiments were approved by the Sunbridge Stewardship District (“District”).

Figure 1 Experimental plots at Sunbridge.

(A) Location of the manipulative experiment in Central Florida (red star) at the (B) Base Camp (welcome center) in Sunbridge. (C) The site also serves as an educational garden with walking trails to promote public access. (D) Experimental plots were arranged in a mirrored split-block design to test the effects of compost addition (C, compost; NC, no compost) and irrigation treatments (R, regular weekly irrigation; AN, irrigation as needed). Map and aerial photo were created with ArcGIS Pro 3.2.2. Plot design courtesy of Cherrylake.

Study design

The experiment was established in January 2022 at the welcome center for the Sunbridge development. It included sixteen 9 × 6 m (55.5 m2) plots planted with the same 26 native plant species (Supplement S1 can be found at doi.org/10.5281/zenodo.15627544) in the exact same layout, and were spaced 1.5 meter apart, where the spaces were either planted with a subset of the experimental plants or mulched with hardwood mulch. Bare spots within the plots were covered with pine straw mulch. The plot designs considered general aesthetics, as they were used for tours and as a native landscape demonstration. The plots included a curving mulched path to facilitate access and accommodate public tours of the experiment (Fig. 1C). The experimental design had four treatment combinations in a factorial, randomized complete block design with four replicates of each treatment combination: regular irrigation with and without compost, and as-needed irrigation with or without compost (Fig. 1D). Soils of eight randomly selected plots were amended with 33 L m−2 LifeSoils COMAND® compost in December 2021, which was mixed into the soil using a hand-operated rototiller (American Honda Motor Co. Inc., Alpharetta, GA, USA) prior to planting at a 15-cm depth. The organic compost had a dry weight composition of 18.87% total carbon and 1.71% total nitrogen, corresponding to a C:N ratio of 11:1. Eight other plots were not treated with compost, but all plots (compost-amended or not) underwent the same tilling process. Plots were irrigated with drip irrigation lines (XFD Dripline, Rain Bird Corporation, Azusa, CA, USA), spaced approximately 30 cm apart. Irrigation was applied equally to all plots for a six-month (January–June 2022) establishment period, but thereafter as two alternative experimental treatments: weekly irrigation based on a traditional irrigation schedule, or as-needed, where plants were only irrigated during periods of extreme drought to keep the plants alive or visually acceptable. “As-needed” plots were irrigated for five days (March 16-20, 2023) during a spring drought and again in summer (August 18-19, 2023) to alleviate hot, dry conditions.

Beetle sampling

Beetles were collected using pitfall traps, adapted from the National Ecological Observatory Network (NEON) pitfall trap protocol (Levan, 2020). The traps were made of 12 oz plastic deli cups (11.4 cm diameter × 7.6 cm deep), with a plastic cover (20 × 20 cm) held in place by four plastic spikes and PVC spacers to leave a 1.5 cm gap between the cover and the top of the container. The cover prevents small vertebrate bycatch and plant litter or other material from blowing into the trap. It shades the trap to reduce evaporative loss and decomposition rate and prevents precipitation accumulation. Pitfall traps were buried flush to the soil level and filled with ∼125 ml of 50% propylene glycol and 50% tap water. To minimize edge effects and reduce disturbance from foot traffic, a 1-meter buffer zone was maintained around the plot boundaries. Two traps were randomly placed in each plot for two consecutive 7-day sampling periods, totaling 14 days of trapping per season during the spring, summer, and fall, over a 16-month period (i.e., March 30-April 13, June 20-July 4, and October 4-18, 2023; March 26-April 9, and June 21-July 5, 2024). Inter-trap distances ranged between 3 to 6 m, providing sufficient distance to assume spatial independence among traps (Snider & Snider, 1986; Ward, New & Yen, 2001). Additionally, we accounted for potential spatial autocorrelation by including trap identity as a random effect in our analyses. Given the slight slope of the plots, we positioned one trap at the higher elevation of the plot and the other at the lower elevation. Traps were oriented along the north-south (N-S) axis in eight plots and along the east–west (E–W) axis in the other eight plots. In the lab, pitfall traps content was filtered through a 0.2 mm sieve to separate the insects from the field preservative solution. The residual material was carefully inspected to ensure no specimens were lost. The specimens were then transferred into Nalgene bottles for initial sorting, and subsequently, beetles were placed into appropriately sized vials and stored in 70% ethanol for further identification and analysis to species or the lowest attainable taxonomic level.

Categorization of beetle functional groups

In this study, we defined functional traits as measurable characteristics that influence beetle performance, ecological role, or response to environmental conditions (Webb et al., 2010). We obtained selected functional traits from the peer-reviewed literature, where beetle traits were chosen based on their hypothesized relationship to the environmental conditions and management practices in these plots. The traits included diet (primary resource used by adults), dispersal ability (wing morphology), size (adult body length in mm), and substrate preference (egg deposition and larva development). Species with overlapping traits (e.g., diet classified as both mycophagous and detritivorous) were assigned to multiple categories to reflect their ecological versatility. Where identification was only possible at a taxonomic level higher than species, or where species-specific information was unavailable, we used the trait typical of the family or subfamily. In cases where information was lacking, personal observations were used. Beetle species list, associated traits and supporting literature are available in Supplement S2 found at doi.org/10.5281/zenodo.15627544.

Abiotic predictors

To determine the effects of soil moisture and temperature on beetles, we measured volumetric water content (VWC, %) using a TDR 350 Soil Moisture Meter (Aurora, IL, USA) at a depth of 12.3 cm, and belowground temperature (BGT, °C) at the same depth using a Digi-Sense WD-20250-93 Thermistor (Vernon Hills, IL, USA). Three readings were taken around each pitfall trap during the morning and early afternoon hours (8:00 to 13:00) on the first day of sampling in week one, the first day of sampling in week two, and the final day of sampling in week two. We calculated weekly averages by combining the first and second readings to represent week one, and the second and third readings to represent week two. Although the thermistor malfunctioned at some sites during the first day of sampling in week two of Summer 2023, we had one valid reading for each week, which was used accordingly. We used mean daily temperatures (TMIN, minimum recorded hourly temperature; TMAX, maximum recorded hourly temperature; TAVG, average hourly recorded temperature) and rainfall data (PRCP) from the Orlando International Airport station (via NOAA’s Global Historical Climatology Network Daily Database; https://www.ncei.noaa.gov/), the nearest weather station to our study site (∼16 km distance), to account for the climatic conditions. From these daily values, we calculated the average conditions for each sampling period by computing the mean temperature and rainfall separately for the first and second week. The comprehensive dataset of abiotic predictors is available in Supplement S3 found at doi.org/10.5281/zenodo.15627544.

Soil physical and nutrition data

To gather soil samples for nutrients, we used a soil probe with a length of 103 cm and a cutting diameter of 1.6 cm to collect five samples 15 cm in depth from each plot: one in the center and four 1.5 m diagonally toward the center and from each corner of the plot. Soil nutrient samples were gathered at the beginning and end of the growing season, March and October 2023. Since we did not anticipate significant changes in the measured soil parameters, and nutrient values within each plot showed minimal seasonal variation, we estimated the June 2023 values by averaging the March and October measurements. For samples used to estimate soil physical properties we used a soil core sampler to collect soil from the top six cm at the end of the growing season in November 2023. Sample locations were randomly assigned within each plot’s top and bottom halves to account for the slope effect. Before sampling, we removed organic material that was recognizable and not fully decomposed: mulch, leaf debris, and roots. After collecting, we stored all samples in a freezer for future analysis.

The UF/IFAS Analytical Research Lab (UF/IFAS ARL) analyzed the nutrient soil samples to determine total carbon (TC), nitrogen (TN), phosphorus (TP), and organic matter (OM) content. Total carbon and nitrogen were measured using the Dumas combustion method, while total phosphorus was assessed by the EPA 365.1 method (EPA, 1983). The organic matter content was determined through loss-on-ignition (Walkley & Black, 1934). The UF/IFAS Agricultural and Biological Engineering Lab estimated soil physical properties, specifically bulk density using methods outlined by Fox et al. (2023). Soil porosity was determined by assuming a particle density of 2.65 g/cm3 and subtracting one from the bulk density divided by particle density. The comprehensive dataset of abiotic predictors is available in Supplement S3 found at doi.org/10.5281/zenodo.15627544.

Vegetation survey

Plant community composition was evaluated every collecting season. A 1 × 1 m quadrat was placed on the ground with the trap in the center and photographed. The photographs were analyzed following an adapted version of the method developed by Peet, Wentworth & White (1998), using cover classes to describe the percentage cover of each species: Class 1 (0%), Class 2 (1–5%), Class 3 (6–25%), Class 4 (26–50%), Class 5 (51–75%), Class 6 (76–95%), Class 7 (96–100%). All native plants were identified to the species level and classified as one of the following morphotypes: shrub (SHR), ground cover forb (GWF), upright forb (UWF), tree (TRE), and grass (GRA). The cover of pine straw mulch (MUL) was also quantified when present. In the results, we report the type of ground cover along with the corresponding class code—for example, GWF7 refers to ground cover forbs, classified as Class 7 (96–100% of coverage). The plant list and seasonal ground coverage are available in Supplement S1 found at doi.org/10.5281/zenodo.15627544.

Data analysis

Taxonomic diversity

We evaluated the experimental effects of vegetation, compost, and irrigation on ground-active beetle communities (as either richness or activity density) using Generalized Linear Mixed Models (GLMMs) with the glmmTMB package in R (Brooks et al., 2017). We modeled beetle richness and activity density as a function of compost as a continuous variable using bulk density (BD) or organic matter (OM) as the surrogate, irrigation (R) as a categorical treatment (either 1 = present or 0 = added as needed), ground cover (native plants and mulch), and abiotic predictors (VWC, BGT, physical and nutritional soil properties, and climatic conditions). Seasonality and spatially-nested blocks, plots, and traps were included as random effects to account for the repeated measures in time and spatial structure of samples. We used the Poisson distribution to model species richness (SR) as it is well-suited for count data. We tried negative binomial distribution (nbinom2) for beetle activity density as it is well-suited for integer data exhibiting overdispersion (Lindén & Mäntyniemi, 2011), but this distribution did not handle over dispersion well. We then log(x + 1) transformed activity density prior to analysis, using Gaussian distribution. Alternative models (including a null) were compared with the Akaike Information Criterion (AICc), and model selection prioritized Akaike weights, which represent conditional probabilities for each model (Wagenmakers & Farrell, 2004). We evaluated GLMM model fit with residual diagnostics, including residual patterns and dispersion, to test the adequacy of our model. We scaled predictors of different units of the best-fitting model to compare their relative importance.

Functional trait structure

We applied a fourth-corner Generalized Linear Latent Variable Model (GLLVM) analysis to assess how beetle species traits were associated with environmental variables in the experiment. We used the gllvm package (Niku et al., 2019; Niku et al., 2024), which integrates three data matrices—environmental factors, species abundance, and species traits—to infer the fourth corner, which reveals trait-environment relationships (Brown et al., 2014).

For this analysis, we selected environmental variables that were most important in the previous GLMM analysis for beetle activity density, along with the selected traits for beetles. We determined the optimal number of latent variables (LVs) for our GLLVM by fitting models with one to three LVs and comparing their AICc values. The model with a single LV (AICc = 16,150.89) outperformed models with two (AICc = 16,280.9) and three LVs (AICc = 16,602.48), indicating that a single latent variable provided the best balance between model complexity and goodness of fit. Therefore, we proceeded with a one-LV model for subsequent analyses. We evaluated GLLVM model fit with residual diagnostics, including residual patterns and dispersion, to test the adequacy of our models. We compared Poisson and negative binomial GLLVMs and chose the negative binomial distribution because it fit better. We assessed the contribution of species traits by comparing model fits for models with trait-environment interactions to those using only environmental predictors. We performed a likelihood ratio test (LRT) using the anova function to compare the goodness of fit of nested GLLVMs and determine whether a more complex model provided better fit for our data. We confirmed that our models met the method’s requirements, with all degrees of freedom differences less than 20 and a sufficient sample size (n = 309). LRTs outputs indicated that models including both environmental variables and trait interactions performed better than those with only environmental predictors (p < 0.0001). Based on this, alternative GLLVMs were informed by GLMM model results and varied in their inclusion of environmental predictors (GWF, MUL, BGT, VWC) and their combined effect with functional traits (diet, wing morphology, body size, and larval substrate). Ground cover classes were transformed into a numeric variable corresponding to the median of the extreme percentage values (i.e., the minimum and maximum values) within each class. Seasonality was included as random effect to address pseudo replication. The nested effect of traps, plots, and blocks were not included because they had the least significance in the GLMM for activity density, and increased model complexity and computational demands without substantially improving model fit. We then ran an AIC comparison to determine the best model among those with trait-environment interaction. For all analyses, we describe results according to Dushoff, Kain & Bolker (2019), where model coefficients with 95% confidence intervals that do not include zero are considered “clear” rather than “significant,” and coefficients of greater magnitude are considered stronger.

Functional diversity metrics

The ‘dbFD’ function in the ‘FD’ R package (Laliberté, Legendre & Shipley, 2014) was used to calculate three functional diversity metrics: functional richness (FRic), functional evenness (FEve), and functional divergence (FDiv). In dbFD, FRic, FEve and FDiv are generally measured as the convex hull volume so we didn’t obtain scores for traps with less than three species.

Functional richness (FRic) quantifies the extent of trait space filled by the species in a community, reflecting the range of ecological roles present, and it usually increases with the number of species. Functional evenness (FEve) describes how evenly species’ abundances are distributed across trait space, increasing when species are more evenly spaced and similarly abundant. Functional divergence (FDiv) measures the degree to which species in a community are spread out within the functional trait space, with a focus on the functional roles of the most abundant species by quantifying functional trait distances from the centroid (or community average) for the dominant species. This metric helps to capture the extent of niche differentiation among species, particularly by considering how the distribution of species’ relative abundances relates to the variability in functional traits. High functional divergence suggests that the community is composed of species with distinct functional roles, potentially leading to greater ecological complementarity and resource partitioning (Mason et al., 2005).

We evaluated models for each of these functional diversity metrics with the glmmTMB package in R (Brooks et al., 2017) using the same structure as models described above for richness and activity density. Season was a random effect to account for repeated sampling. We specified a beta family distribution because it works well with response variables that are continuous and constrained between 0 and 1 (Ferrari & Cribari-Neto, 2004). Alternative models (including a null) were compared with the AICc, and we scaled predictors of different units of the most parsimonious model to compare their relative importance. We assessed model fit with residual diagnostics, including residual patterns and dispersion, to test the adequacy of our model.

All analyses were conducted using R version 4.4.3 (R Core Team, 2025). Methodological details are included in Supplement S4 and datasets along with metadata in Supplement S5; both available at doi.org/10.5281/zenodo.15627544.

Results

In total, we collected 10,546 beetles of 107 different taxa. We refer here to beetle species richness because 73% of the beetles were identified to that level. The remainder (27%) were identified to genus or subfamily labeled as morphospecies (e.g., genus and species A).

Of the taxa collected, 86 species (80.4%) were native and 21 (19.6%) were non-native, most of Neotropical origin. The native fauna was dominated by Xyleborus affinis Eichhoff, 1868 (Curculionidae: Scolytinae) and Rismethus squamiger (Champion, 1894) (Elateridae: Agrypninae), which accounted for 38.1% and 23.3% of all beetles, respectively, together representing 61.6% of the total catch. The next three most abundant native species were Ahasverus rectus LeConte 1854 (Silvanidae: Silvaninae), Anotylus insignitus (Gravenhorst, 1806) (Staphylinidae: Oxytelinae), and one undetermined species of Aleocharinae (sp. A) (Staphylinidae), comprising 5.9%, 2.2%, and 2.1%, respectively. Among non-native species, Heteroderes amplicollis (Gyllenhal, 1817) (Elateridae: Agrypninae) and Gondwanocrypticus platensis (Fairmaire, 1884) (Tenebrionidae: Diaperinae) were the most abundant, accounting for 7.5% and 4.3% of all beetles, together representing 65% of non-native individuals (41.3% and 23.7%, respectively). Tetragonoderus laevigatus Chaudoir, 1876 (Carabidae: Harpalinae), which appears to have recently expanded its range into Central Florida (Pandolfi & Schnepp, 2024), represented 16% of non-native beetles, making it the third most abundant non-native species, followed by Poecylocrypticus formicophilus Gebien, 1928 (Tenebrionidae: Diaperinae) and Coccotrypes carpophagus (Hornung, 1842) (Curculionidae: Scolytinae), which comprised 6.2% and 5.0% of non-native taxa, respectively.

Notably, two species recently identified as new to the United States were also collected in this study: Mimogonus fumator (Fauvel, 1889) (Staphylinidae: Osoriinae), representing 0.42% of non-native taxa, and Vacusus parvus (Pic, 1910) (Anthicidae: Anthicinae), representing 0.05% of non-native taxa (Pandolfi et al., 2024; Pandolfi & Chandler, 2025). See Supplement S2 (available at doi.org/10.5281/zenodo.15627544 for additional information.

Taxonomic diversity

Species richness was most effectively predicted by the additive effects of irrigation, soil bulk density, GWFs, belowground temperature, a quadratic effect of moisture, and the random effects of season and block/plot/trap (AICc = 1559.8, Weight = 0.5512, R2 = 0.55). In that model, irrigation did not clearly impact beetle richness (β = 0.48, p = 0.60) and bulk density had only a marginally significant negative effect on beetle richness (β = −0.08, p = 0.06). High levels of GWFs clearly and positively affected beetle richness (GWF5: β = 0.413, p = 0.022; GWF6: β = 0.439, p = 0.018; GWF7: β = 0.528, p = 0.008). Belowground temperature had a significant positive effect (β = 0.29, p = 2.38e−05), while soil moisture content exhibited a clear quadratic relationship with beetle richness (N ∼0.077*moisture −0.109*moisture, with a significant negative quadratic term (p = 3.68e−07), while the linear term was not statistically significant (p = 0.12) (Fig. 2A).

Figure 2 Significant covariates obtained from Generalized Linear Mixed Models (GLMMs).

(A) Species Richness. (B) Activity Density. BGT, belowground temperature (°C); VWC, volumetric water content (%); GWF, ground cover forb (class); MUL, pine straw mulch (class).

Beetle activity density was most efficiently modeled with additive effects of irrigation, bulk density, pine straw mulch, belowground temperature, quadratic moisture, and random effects of season and block/plot/trap (AICc = 808.7, Weight = 0.46, R2 = 0.61). Irrigation and bulk density did not clearly impact beetle activity density (β = 0.19, p = 0.317 and β = −0.12, p = 0.215, respectively). Mulch showed a negative trend with beetle activity density, with a significant reduction observed for mulch cover class 3 (MUL3: β = −0.325, p = 0.049) and a marginally significant effect for class 6 (MUL6: β = −0.537, p = 0.06). Belowground temperature had a clear, positive effect (β = 0.274, p = 0.024), and beetle activity density peaked at intermediate soil moisture content (N ∼0.34*moisture −0.153*moisture, p = 0.0001 for linear coefficient and p = 1.92e−06 for the quadratic coefficient) (Fig. 2B).

Functional trait structure

The beetle assemblage was dominated by species that were small in size, dimorphic or macropterous. Most taxa presented selection for dead plant material or soil as larval substrate, and mycophagous and phytophagous feeding strategies were most common. Conversely, dung beetles—those that feed on and lay eggs in dung—were rarely encountered (Fig. 3).

Figure 3 Activity Density patterns of ground-active beetles in relation to functional traits.

(A) Adult body size (mm). (B) Wing morphology (BRA, brachypterous; DIM, dimorphic; MAC, macropterous). (C) Larval substrate (DNG, dung; DPL, dead plant material; DPL+LPL, dead and live plant material; FRT, decaying fruit; LPL, live plant material; SOI, soil; SOI+DNG, soil and dung; SOI+DPL, soil and dead plant material; SOI+LPL, soil and live plant material). (D) Diet (COP, coprophagy; DET, detritivory; COP+DET, coprophagy and detritivory; MYC, mycophagy; MYC+DET, mycophagy and detritivory; PHY, phytophagy; POL, polyphagy; PRD, predatory).

Trait coefficients from the fourth-corner GLLVMs indicated that larger-bodied species were positively associated with wetter soils (β = 0.24, p = 0.0017) and negatively correlated with increased belowground temperatures (β = −0.60, p = 3.18e−07). All diet types, except mycophagy, were positively associated with high belowground temperatures, and beetles with a detritivore diet were negatively associated with high soil moisture (β = −1.06, p = 0.022). Beetles that use live plants for their egg and larvae development were negatively associated with high belowground temperatures (β = −1.53, p = 0.027). Finally, beetles with dimorphic wings (wing development different in the two sexes) were positively associated with warmer soils (β = 0.95, p = 0.049). Although ground cover covariates (GWF and MUL) were retained in the best-fitting GLLVM based on model selection criteria, they did not show statistically significant associations with individual traits. Only predator beetles showed a marginally negative correlation with mulch (β = −1.029, p = 0.08, Fig. 4).

Figure 4 Trait coefficients derived from the Generalized Linear Latent Variable Models (GLLVMs).

Warm colors show positive relationships while cool colors show negative relationships. Model coefficients with 95% confidence intervals that do not include zero and have p < 0.05 are marked with an asterisk. Species traits: Diet: POL, polyphagy; MYC, mycophagy; PHY, phytophagy; PRD, predatory; DET, detritivory; COP, coprophagy; COP+DET, coprophagy + detritivory; MYC+DET, mycophagy + detritivory. Larval substrate: SOI, soil; LPL, live plant; SOI+DPL, soil + dead plant; DPL+LPL, dead plant + live plant; SOI+LPL, soil + live plant; FRT, decaying fruit; DNG, dung; SOI+DNG, soil + dung. Wings: MAC, macropterous; DIM, dimorphic. Environmental Variables: BGT, belowground temperature; VWC, volumetric water content; GWF, ground cover forb; MUL, pine straw mulch.

Functional diversity metrics

Functional richness (FRic) increased with species richness (SR), though the rate of increase declined at higher levels of SR (Fig. 5A). Functional evenness (FEve) showed a slight decrease as SR increased, consistent with increasing trait redundancy (Fig. 5B), and functional divergence (FDiv) increased with SR, consistent with more trait extremes being included in richer communities (Fig. 5C).

Figure 5 Relationship between Species Richness and Functional Diversity metrics.

(A) Functional Richness (FRic). (B) Functional Evenness (FEve). (C) Functional Divergence (FDiv). Points represent individual pitfall traps used in this study.

FRic was most effectively predicted by bulk density (BD), irrigation (R), mulch cover (MUL), belowground temperature (BGT), quadratic soil moisture (VWC) and seasons included as random effect. The model explained a moderate proportion of the variance in FRic (AICc = −547.2, Weight = 0.1062, R2 = 0.51). Irrigation (R) showed a marginally negative correlation with FRic (β = −0.15, p = 0.056). FRic significantly decreased in mulch cover class 3 (MUL3: β = −0.342, p = 0.008), class 4 (MUL4: β = −0.349, p = 0.004), and class 5 (MUL5: β = −0.533, p = 0.012), and significantly increased with belowground temperature (β = 0.306, p = 1.21e−10), and showed a significant quadratic response to soil moisture (VWC) (β = −0.089, p = 0.004).

FEve was most effectively predicted by bulk density (BD), irrigation (R), precipitation (PRCP), lowest hourly temperature (TMIN), and highest hourly temperature (TMAX), with season included as a random effect. The model explained a substantial proportion of the variance in FEve (AICc = −259.2, Weight = 0.1049, R2 = 0.68). Precipitation and the lowest temperatures showed a significant negative effect on FEve (β = −0.155, p = 0.018 and β = −0.194, p = 0.043, respectively).

FDiv was most effectively predicted by organic matter (OM), irrigation (R), precipitation (PRCP), lowest hourly temperature (TMIN), and highest hourly temperature (TMAX), with season included as a random effect. The model explained a very high proportion of the variance in FDiv (AICc = −495.4, Weight = 0.1326, R2 = 0.92). Precipitation and the highest temperatures showed a significant positive effect on FDiv (β = 0.261, p = 7.77e−05 and β = 0.320, p = 0.000479, respectively), while minimum temperatures showed a significant negative effect (β = −0.177, p = 0.044).

Discussion

Urban areas occupy less than 3% of the Earth’s surface but account for 60% of residential water consumption (Grimm et al., 2008). In Central Florida, residential landscape irrigation represents the majority form of household water usage (Landon et al., 2018; Sadalla et al., 2014). In this context, implementing effective water conservation strategies is crucial, especially those that reduce water usage in outdoor amenity landscapes (Martin, 2015). While drip irrigation systems offer benefits relative to the common spray irrigation (Jarwar et al., 2019), another effective approach is the selection of native, drought-tolerant plants adapted to the local conditions, such as those used in our study (Martin, 2015; Shapiro et al., 2015; Alam et al., 2017). Moreover, soil quality is a key factor influencing landscape sustainability. Suburban soils often suffer from compaction, low porosity and reduced organic material content (Evanylo et al., 2016). Incorporating compost into these soils can reduce bulk density, improve plant health, and enhance net primary productivity (De Lucia et al., 2013; Layman et al., 2016). Soil amendments may also contribute to pest suppression by promoting beneficial predators (Bell et al., 2008) and increasing the abundance of soil-dwelling arthropod populations (Mathews, Bottrell & Brown, 2002; Brown & Tworkoski, 2004). These soil communities are vital to ecosystem functioning, but their responses to urbanization remain largely unknown due to the complex and taxon-specific effects of urban landscape transformation (Magura, Lövei & Tóthmérész, 2010; Baldock et al., 2015; Knop, 2016; Fenoglio, Rossetti & Videla, 2020). While much of the existing literature focuses on carabid beetles—demonstrating clear responses to vegetation management, landscape modification, and urban development (e.g., Martinson & Raupp, 2013; Niemelä et al., 2002; Magura & Lövei, 2021)—broader beetle community responses, especially in relation to habitat structure and microclimatic conditions, are still poorly understood (Yong et al., 2020), particularly in urban and suburban environments.

Our study aimed to address this gap by examining the complex interactions between local habitat and microclimatic factors affecting soil-active beetles, in an urban context. The significant positive impact of native forbs on pollinators, such as wild bees, has been widely demonstrated (Williams et al., 2015; Williams et al., 2024), but their effect on soil active beetles remains largely unexplored. Our findings of a strong positive association between species richness and ground cover forbs suggests that increased habitat complexity and resource availability—likely through the provision of shelter, foraging opportunities, and suitable microclimatic conditions—may support more diverse soil-active beetle communities. Conversely, the negative relationship between pine straw mulch coverage and beetle activity density suggests that excessive ground coverage with pine straw mulch may alter soil conditions in ways that limit beetle movement or resource availability. Mulching has been shown to negatively impact ground-nesting wasps (Georgi et al., 2022) and suppress soil arthropods (Burkhard, Lynch & Percival, 2008). Pine straw mulch can reach high surface temperatures on sunny days and, due to its insulating properties, may create thermally stressful conditions for ground-dwelling beetles (Renkema et al., 2011). Although effective at suppressing weed competition, mulch may reduce beetle habitat suitability by altering ground-level conditions. It may also increase flooding risk during heavy rains and raise beetle mortality through shallower retreats and possible more frequent encounters with ground predators (Renkema et al., 2011).

As expected, soil temperature and moisture were significant local drivers of beetle activity density and richness. Belowground temperature emerged as a key, positive predictor of both metrics. Within natural limits, ectothermic organisms, like beetles, respond positively to warmer conditions due to greater metabolic activity, faster development rates, and greater reproductive success (Harvey et al., 2023). In our study, although irrigation was used as a binary predictor (1 = present, 0 = added as needed), the continuous measure of volumetric water content proved to be more informative for soil-active fauna. In fact, while irrigation did not significantly affect beetle communities, soil moisture at a 12 cm depth played an important role in explaining beetle community patterns. We found clear quadratic relationships between soil moisture and both adult beetle species richness and activity density, suggesting that moderate moisture levels are beneficial. However, excessive moisture likely creates unfavorable conditions—such as soil saturation or reduced oxygen availability—that inhibit beetle activity (Keenan et al., 2018). This pattern is consistent with recent findings from tropical regions (Newell, Ausprey & Robinson, 2023) and helps explain the seasonal dynamics of ground-dwelling beetles. This result highlights the importance of managing irrigation to prevent unnecessary overwatering, which depletes natural resources and creates unfavorable conditions for soil fauna. Interestingly, bulk density showed a marginal negative trend with species richness. Compost incorporation into soils should reduce soil bulk density and possibly increase habitat suitability for ground-active beetles. The converse process—compaction of the soil—can limit burrowing opportunities and alter soil aeration, which may disproportionately affect species reliant on subterranean activities (Müller et al., 2022). We expected a clearer effect of bulk density or organic matter on beetles in the plots, but other factors had a more predominant role. This finding aligns with recent research on soil amendments in a residential development, which reported no detectable effects on ground-dwelling invertebrates within the first year following disturbance, with positive impacts only emerging in the second year, suggesting that the benefits of compost may take longer to become evident (Borden et al., 2022).

While traditional metrics like species richness and activity density provide important insights, our findings show that functional diversity capture distinct ecological patterns influenced by a combination of soil, microclimate, and short-term climatic conditions emphasizing the importance of incorporating functional traits in urban ecology studies (Kotze et al., 2011; Pey et al., 2014). Our analyses of the relationship between environmental factors and species traits offer correlative insights into potential ecological drivers of beetle communities, though further research is needed to establish direct mechanistic links. Larger beetles showed a positive correlation with wetter and cooler soils. These soils are more thermally stable, which may benefit larger-bodied beetles that may be more sensitive to overheating. These results align with previous studies where the mean body mass of invertebrates and other organisms decreased with rising temperatures, suggesting that smaller species may become more prevalent as temperatures increase (Daufresne, Lengfellner & Sommer, 2009; Robinson et al., 2018). The negative correlation between species that need live plant material (i.e., endophytic or exophytic plant tissues) for their eggs and larval development with warmer soil conditions may be due to the physiological stress that higher temperatures impose on the plants, potentially reducing habitat suitability and resource availability for the developing larvae. This aligns with findings from Warner et al. (2021) that suggest that higher soil temperatures can alter plant-invertebrate interactions, potentially affecting species that depend on live plants for reproduction and development. All diet types were positively associated with increased belowground temperatures; only mycophagous beetles did not show a significant relationship. Detritivore beetles showed a negative correlation with wetter soils. While increased soil moisture promoted decomposition in colder, drier ecosystems such as those studied by Robinson et al. (1995), the effects of moisture on decomposition are highly context dependent. A study on soil communities in a temperate agricultural field reported that dryness increased consumer pressure on microbial resources (fungi and bacteria) (Lang et al., 2014). In warm, humid subtropical environments like Florida, high soil moisture can lead to saturation, reducing oxygen availability (Zausig, Stepniewski & Horn, 1993), which may favor certain types of microbial decomposers (e.g., anaerobic bacteria) over others. This shift can alter the decomposition pathway and make the substrate less palatable (Kaur et al., 2020) which may in turn lead to lower presence of detritivore beetles. Finally, beetles with dimorphic wings were more common in areas with warmer belowground temperatures. Since the group classified as dimorphic primarily included bark beetles (Curculionidae: Scolytinae), in which males are typically wingless (Sobel, Lucky & Hulcr, 2018), this pattern may reflect enhanced reproductive success or increased male surface activity under warmer soil conditions. The absence of any significant association between ground cover and functional traits may be due to the influence overlaps with stronger predictors like belowground temperature or soil moisture, which showed significant associations with multiple traits. These results suggest that forbs and mulch cover may have a more subtle or diffuse impact on trait filtering in beetle communities, rather than strongly shaping specific ecological traits.

Functional diversity metrics revealed additional patterns. The decelerating increase of functional richness (FRic) with species richness (SR) aligns with the scenarios proposed by Micheli & Halpern (2005), where additional species contribute less novel trait space at higher richness levels, eventually reaching an asymptote, potentially due to trait saturation or the presence of functionally similar species. The reduction and constraint in functional evenness (FEve) with increasing SR have already been observed in other studies (e.g., Li et al., 2022) and may reflect trait redundancy or non-random environmental filtering in richer communities. The increase and stabilization of functional divergence (FDiv) with SR indicates that species with extreme trait values are more consistently present in species-rich communities (Mason et al., 2005). These broad patterns were further refined by environmental characteristics, with some factors driving trait-specific filtering. Functional richness (FRic) declined in the presence of higher mulch cover, suggesting that excessive ground cover may limit the types of taxa adapted to those environmental conditions and the range of functional traits present. This conjecture aligns with a study on rainforest restoration that demonstrated how deeper organic mulch did not create more suitable conditions for various faunal arthropods (Nakamura et al., 2009). In contrast, FRic increased with higher belowground temperatures and showed a quadratic relationship with soil moisture, indicating that moderate moisture levels and temperatures promote greater functional trait diversity. Experimental manipulations of soil moisture have shown the same non-linear pattern on some soil microarthropods, like oribatid mites and Collembola, which exhibit markedly distinct functional traits (Tsiafouli et al., 2005). Functional evenness (FEve) decreased with increasing precipitation and with the minimum recorded hourly temperature, implying that wetter and cooler conditions lead to dominance by certain trait combinations, reducing trait balance. Meanwhile, functional divergence (FDiv) increased with greater precipitation and with the maximum recorded hourly temperatures but declined with minimum recorded hourly temperatures. Our results suggest that more extreme conditions, in terms of precipitation and heat, promote functionally distinct species. Conversely, milder temperatures and less intense precipitation may favor more functionally similar assemblages. These findings suggest non-random community assembly driven by strong environmental filtering, where only functionally unique or stress-tolerant species can persist (Boucek & Rehage, 2014). This process likely contributes to consistent patterns of trait composition across spatial scales, as demonstrated in recent studies (e.g., Lamanna et al., 2014). Overall, our findings indicate that functional richness (FRic) is primarily driven by soil conditions, which provide the foundation for supporting a broader range of traits. In contrast, short-term climatic conditions such as precipitation and temperature influence the distribution of these traits, indicating that species presence is tied to stable soil features, while their relative abundances fluctuate in response to changing weather conditions. The opposing effects of precipitation on functional evenness (FEve) and functional divergence (FDiv) indicate that precipitation favors certain traits, reducing trait balance, but increasing overall spread. This means that even if only a few traits dominate, those traits are more divergent from the average (traits are more different from each other).

The presence of non-native beetles in our experimental plots—accounting for nearly 20% of all recorded taxa—adds an additional layer of complexity and underscores the role of urban and managed landscapes as hotspots for adventive species. The predominance of Neotropical-origin species among the non-native taxa suggests that regional climatic conditions may facilitate their establishment in Florida. Notably, the two species detected in this study, representing recent introductions to the United States (Pandolfi et al., 2024; Pandolfi & Chandler, 2025), underscore the need to monitor species composition in urbanized landscapes and assess the effects of urban development on insect community dynamics (Gaertner et al., 2017). Non-native species are not inherently harmful but may instead represent biological adaptability for communities to changing conditions. They may fill ecological roles that have been diminished by habitat loss, contribute to novel food webs, or even facilitate certain ecosystem functions, particularly in urban areas where the landscape itself is a fusion of introduced and restructured elements (Schlaepfer, Sax & Olden, 2012). However, it is important to recognize that all biological invasions begin with species introductions, and some non-native insects have become highly disruptive invaders, causing ecological, economic, or health-related harm (Fortuna et al., 2022).

Despite the limited scale of the experiment, our results demonstrate that landscape management strategies influence microclimate and soil conditions, which in turn affect insect communities. This suggests that even fine-scale habitat modifications within residential lots, such as changes in vegetation structure and soil moisture, can strongly influence ground-dwelling beetles. The Orlando metropolitan area is one of the most rapidly developing regions in the US and serves as a prominent example of conventional residential development where sustainable landscaping practices at the neighborhood level may offer ecological benefits. Future research should examine how treatment scale and spatial arrangement shape insect responses across urban areas.

Concluding remarks

Our findings underscore the need for holistic approaches to studying how urban development shapes insect biodiversity, as they reveal complex interactions between local environmental factors and community structure. Despite their ecological importance and global diversity, beetles have received comparatively little conservation attention in urban settings. In our study, their sensitivity to both fine-scale habitat features (e.g., ground cover and soil conditions) and short-term climatic fluctuations (e.g., precipitation and temperature extremes) highlight their potential as valuable indicators of how urban development affects insect community composition and diversity. Previous studies have shown that urban green spaces select certain traits, such as flightless, smaller, and habitat-generalist species. By examining functional diversity, we gained insights into the selective pressures of local-scale ecological filters shaping beetle functional traits, and how these impact their presence in a newly established residential development. This research offers new insights into the mechanisms through which landscape design and soils management can shape ecological communities. Understanding these dynamics is critical for fostering more sustainable and biodiversity-friendly urban spaces.

The authors acknowledge Tavistock Development Company for their overall support of this project, and Cherrylake, Inc. and LifeSoils LLC for their valuable contributions to the design and maintenance of the experimental plots. The authors also acknowledge the Florida Department of Agriculture and Consumer Services, Division of Plant Industry, for their support of this work. Special thanks go to Sandor Kelly for granting access to the University of Central Florida Collection of Arthropods (UCFC, Bug Closet), and to all the dedicated students who assisted with fieldwork.

Additional Information and Declarations

Competing Interests

Author Contributions

Field Study Permissions

Data Availability

The authors declare there are no competing interests.

Alessandra Pandolfi conceived and designed the experiments, performed the experiments, analyzed the data, prepared figures and/or tables, authored or reviewed drafts of the article, and approved the final draft.

Patrick J. Bohlen conceived and designed the experiments, authored or reviewed drafts of the article, and approved the final draft.

Basil V. Iannone III conceived and designed the experiments, authored or reviewed drafts of the article, and approved the final draft.

Brooke L. Moffis conceived and designed the experiments, performed the experiments, authored or reviewed drafts of the article, and approved the final draft.

Paul E. Skelley analyzed the data, authored or reviewed drafts of the article, and approved the final draft.

David G. Jenkins analyzed the data, authored or reviewed drafts of the article, and approved the final draft.

The following information was supplied relating to field study approvals (i.e., approving body and any reference numbers):

Field experiments were approved by the Sunbridge Stewardship District (“District”).

The following information was supplied regarding data availability:

The plant list and seasonal ground coverage (Supplement S1), beetle species list, associated traits and supporting literature (Supplement S2), comprehensive dataset of abiotic predictors (Supplement S3), model covariates and statistical outputs (Supplement S4), complete datasets including metadata (Supplement S5), and code are publicly available at Zenodo: Pandolfi, A. (2025) Supplementary material for Pandolfi et al. (2025) Experimental assessment of the effects of sustainable landscaping practices on beetles in a new residential land development [Data and code archive]. Zenodo. https://doi.org/10.5281/zenodo.15627544.

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
