# Peer review of "Experimental assessment of the effects of sustainable landscaping practices on beetles in a new residential land development"

_PeerJ, doi:10.7717/peerj.20415_

## Round 0.1 · original submission · Major Revisions

· Academic Editor

Major Revisions

Dear Dr. Pandolfi,

After this first review round, I believe your manuscript may be accepted for publication after major reviews indicated by the reviewers are considered. Both reviewers indicated that your manuscript was well written. Still, one of them indicated major reviews are needed to increase the strength of your study.

Sincerely,

Daniel Silva

Reviewer 1 ·

Basic reporting

Very well written paper. It was a pleasure to read.
Nice use of references throughout. A few suggestions for additional references to include.
Great figures. A few minor edits needed.

Experimental design

Great experimental design.
Relatively low replication of experimental treatments (only 4), but were still able to detect significant treatment effects.
Solid methodological descriptions, only a few minor suggestions.
Really like the statistical approaches selected.

Validity of the findings

Findings are supported by the data, experimental design, and selected analyses.

Additional comments

I found this paper enjoyable to review. Overall, very well written and nicely presented. I have several minor suggestions that are shared in the attached file.
The biggest areas for improvement, in my opinion, are tying in more of the relevant literature related to ground beetle communities in urban landscapes and the effects of some of the treatments that were evaluated. I have made some suggestions in the attachment.
Great work!

Annotated reviews are not available for download in order to protect the identity of reviewers who chose to remain anonymous.

Reviewer 2 ·

Basic reporting

a. Clear and unambiguous, professional English used throughout.
The article has several native-speaker authors, making it clear, unambiguous, and technically accurate. It adheres to professional standards of courtesy and expression.
b. Literature references, sufficient field background/context provided.
The article shows many recent references. However, several parts of the text need better substantiation and discussion, especially in the discussion section, which will be addressed later. More references are necessary to explore the topic more thoroughly.
c. Professional article structure, figures, tables. Raw data shared.
The structure is largely in line with the journal's standards, but the conclusion is too long, seeming like a displaced part of the discussion. As will be suggested later, the conclusion should be simplified, and the references and text from this topic should be used to enrich the discussion section.
Figures are relevant to the article's content, of sufficient resolution, and are appropriately described and labeled.
It is necessary to include a table with the captured species and relevant information about each of them. A table with basic data like this for the species or environmental data obtained was not provided.
The analysis and results are relevant to the hypotheses.

Experimental design

According to the authors and a particular research that I did, this article seems to be novel in microclimatic studies of the effect of urban expansion on soil insects. The authors used a significant number of sampling units, and the analytical methods are adequate to answer the proposed questions.
One thing that remains unclear is whether the distance between sampling points (1.5 m) would be sufficient to establish sampling independence. The authors should justify whether this distance constitutes sampling independence and provide references that support their decision. Furthermore, it would be helpful to have a diagram or illustration of the trap distribution at each sampling point and a photo or illustrative diagram of the traps used.
The categorization of functional groups needs to be better explained. For example, what do the authors consider "wing morphology"? The discussion mentions dimorphic wings. What do the authors mean by this? They should enumerate or better describe what they considered "traits" in their analysis, that is, what specific traits they analyzed.

Validity of the findings

The study is impactful because it demonstrates that urban expansion must be conducted with consideration for the negative impacts that may occur on flora and fauna, without neglecting soil invertebrates, which are so important for nutrient cycling and soil quality. The study demonstrated the impact of irrigation, soil cover, soil moisture, and nutrient structure and quality on soil entomofauna and suggested better approaches and management to mitigate these impacts.
The article is also novel because, according to the authors, there are no studies on the impact of urban expansion on soil entomofauna, especially beetles.
The sampling design needs to be better described, with more details on trap distribution, spacing between traps within the sampling point, and sample independence.
The analyses are robust and adequate to meet the proposed objective.
Baseline data, such as species list, sample and recording station, abundance, are not presented. This makes it difficult to compare the results of this study with other locations. The conclusions seem more like part of the discussion and should be redistributed there.

Additional comments

The authors should better explain the negative relationship between detritivores and soil moisture in the discussion, including more references, starting from line 486.
The discussion section, in general, is poorly developed and contains few references and comparative or explanatory examples. This is particularly true when the authors discuss FEve and FDiv, in the paragraph starting at line 534. There is no reference in the entire discussion between lines 541 and 561. There is no more in-depth explanation of the relationships presented. The discussion should be further developed and enriched with other works and references. The conclusions in this section, especially between lines 541 and 561, are shallow assumptions and are not supported, corroborated, or contrasted with other results found in the literature. Are there no other studies on the topic, even if with other organisms or conditions? These results need to be better discussed and explained.
The list of beetle species sampled should be presented, as already noted, to provide an understanding of the most frequently encountered species and groups and the conditions under which each species was found. The paragraph starting at line 563 discusses Neotropical species and new records for the studied region, but does not present the specific species, either in a table or described in the text. A table should be presented indicating the species name, trap, abundance, whether native or not, classifications according to the traits described in the material and methods, the vegetation classes in which they occur, and all other basic information about the species used for the analyses. There is no information on which groups were more abundant or rarer, nor the relationship of taxonomy to environmental characteristics, such as species more or less adapted to different evaluated characteristics. Furthermore, this data on species nomenclature and characteristics should be used in the discussion, enriching the topic, such as the species mentioned but not named in the paragraph starting at line 563.
The conclusions presented in the concluding remarks section fit better into the discussion, fleshing it out and enriching it, justifying and corroborating, or contrasting the results. I suggest combining the discussion with the text presented in the concluding remarks section, distributing the statements from the latter in the appropriate places. In the section they are currently in, they are disjointed and explain nothing. Such an adjustment would certainly bring greater depth and credibility to the results and the discussion. The conclusion itself is closer to what is presented from line 612 onward.

---

## Round 0.2 · Minor Revisions

· Academic Editor

Minor Revisions

Dear Dr. Pandolfi,

Following this new review round, both reviewers believe that your manuscript is nearly suitable for publication in PeerJ after minor revisions. I commend the authors for their efforts in improving their manuscript. I also thank the reviewers for their valuable input and suggestions on the revised manuscript. As soon as the minor changes are implemented.

Sincerely, Daniel Silva

Reviewer 2 ·

Basic reporting

The requested changes were made.

My considerations about some tables and information that should be included in the article were responded to by informing me that they were part of supplementary material that was not made available on the submission platform, in a place designated for this purpose. This supplementary material was uploaded to an external repository, but there was no mention of it in the initially submitted text. The authors provided the repository link in the review response, but the text still fails to mention either the supplementary material or the repository. The existence and indication of the supplementary material should be in the text; that is, the reference to the supplementary material should be in the main text, for example, the first paragraph of the results should indicate the reference to supplementary material S2.

Experimental design

The requested changes were made.

The "data analysis" topic only lists the program used to run the analyses, but in fact, it is a subtopic of materials and methods, and the topics subsequent to data analysis are subtopics of that. Therefore, this sentence about using R should be removed from there and placed at the end of the presentation of all the data analyses performed.

Validity of the findings

The requested changes were made.

Additional comments

The conclusion, as the name suggests, concludes the paper and should present the authors' conclusive considerations on the findings of the study. Therefore, it should not contain citations. This makes it appear as a mini-discussion. Citations should be removed from the conclusion.

·

Basic reporting

As this study has already been reviewed, I have only minor further comments on basic reporting, detailed in the Additional Comments section. The authors present a well-designed study examining patterns of beetle richness, diversity and activity in a suburban development. They provide convincing evidence that sustainable landscaping practices in urbanized environments are beneficial in supporting Coleoptera.

Experimental design

No further comments, the authors have thoroughly addressed reviewer concerns regarding study design.

Validity of the findings

This article contributes important information to our body of ecological knowledge of a relatively understudied taxon, and of especial importance, lends support to the idea that built communities should fit into local contexts and must provide for ecological function and biodiversity maintenance in a rapidly urbanizing world. I particularly commend anyone who takes on Coleoptera as a target taxon, as this is a notoriously difficult group of organisms.

Additional comments

After careful consideration, I feel that the authors have adequately addressed comments provided by reviewers and recommend this article for publication. In particular, their revision of the discussion has rendered this section much improved. However, I noticed a few additional minor issues, mostly editorial in nature, which I have detailed in line numbers below. These line numbers match with the revised tracked changes document sent to me.

Line 133 The first sentence of this paragraph seems to be a run-on. Consider revising to: “One promising solution to these issues is incorporating native plants into suburban landscapes. Native plants are adapted to local climate conditions, resulting in lower management needs such as reduced irrigation demand compared the non-native plants frequently used in these settings.”

Line 150 Citation needed for poor quality/management requirements of urban soils.

Line 333 The phrase “as evidenced by other studies” is extraneous as you cite these other studies immediate after.

Line 776 I’m not sure that Lang, 2014 counts as a “recent” study, given the fact that 2014 was over a decade ago, as unbelievable as this may seem.

Supplementary Materials: I did not receive SM for this manuscript beyond residual diagnostics and a research agreement, both marked “not for publication”. It seems from some reviewer comments that this may have been the case for them as well. For example, the functional trait categorization questions may have been avoided. Similarly, I also wanted to check out the plant list from line 290. The supplements specifically referred to in the rebuttal aren’t cited in the text. I couldn’t actually figure out from briefly looking at the submission guidelines how PeerJ prefers for SM to be referenced in the text. Something to check on with editorial, which might clear up confusion for readers.

---

## Round 0.3 · accepted · Accept

· Academic Editor

Accept

Dear Dr. Pandolfi,

After this new review round, I am pleased to accept your manuscript for publication in PeerJ! Congratulations to you and your co-authors for the acceptance!

Sincerely,
Daniel Silva

Reviewer 2 ·

Basic reporting

All requests previously made have been met.

Experimental design

Correction requests were made, improving the quality of the description of the material and methods.

Validity of the findings

All requests were met, improving the presentation of results and understanding of findings.

Additional comments

All requests previously made have been met.

·

Basic reporting

The authors have made the requested changes.

Experimental design

I had no comments on experimental design since they had been addressed prior to my review.

Validity of the findings

The authors have made the requested changes.

Additional comments

The authors have made the requested changes.